# Influence of Palliative Care Training on Nurses’ Attitudes towards End-of-Life Care during the COVID-19 Pandemic in Spain

**DOI:** 10.3390/ijerph182111249

**Published:** 2021-10-26

**Authors:** Encarna Chisbert-Alapont, Isidro García-Salvador, María Jesús De La Ossa-Sendra, Esperanza Begoña García-Navarro, Marisa De La Rica-Escuín

**Affiliations:** 1Day Hospital, La Fe University and Polytechnic Hospital, 46026 Valencia, Spain; encarna.ch7@gmail.com; 2Research Group of the Spanish Nursing Association of Palliative Care AECPAL, 28036 Madrid, Spain; isidro.gs@hotmail.com (I.G.-S.); mariajesusossa@cudeca.org (M.J.D.L.O.-S.); marisadlrscn@hotmail.com (M.D.L.R.-E.); 3Oncology Service, Dr. Peset Hospital, 46017 Valencia, Spain; 4Cudeca Foundation, Institute of Biomedical Research of Malaga (IBIMA), 29631 Benalmádena, Spain; 5ESEIS Research Group, Department of Nursing, University of Huelva, 21007 Huelva, Spain; 6Institute of Health Research of Aragon, University of Zaragoza, 50009 Zaragoza, Spain

**Keywords:** palliative care, end-of-life care, nursing education, nursing training, COVID-19

## Abstract

Aim: This study aims to assess the influence of training on nurses’ attitudes toward end-of-life care during the COVID-19 pandemic alarm state in Spain. Design: Cross-sectional descriptive study. Data collection was carried out by means of an ad hoc questionnaire using Google Forms in April and May 2020. The score of attitudes toward end-of-life care was used, to which sociodemographic variables and training in palliative care were added. Methods: Data were collected from 238 nursing professionals who had cared for COVID-19 and non-COVID-19 adult patients at the end-of-life stage in a hospital or nursing home. Results: Results showed that 51% of the nurses in the sample had training in palliative care. However, the percentage decreased to 38.5% among those who cared for COVID-19 patients and to 44.5% in those who cared for non-COVID-19 patients. In relation to attitudes about end-of-life care, more positive attitudes and a higher mean score were found in the trained group. Conclusions: Palliative care training is a key element in end-of-life care and is even more important in times of COVID-19. Impact: Although end-of-life accompaniment has been studied, few studies have included the influence of training on this during the pandemic. This study identifies key elements of accompaniment and training in a comparison of COVID-19 and non-COVID-19 patients during the pandemic. In relation to attitudes toward end-of-life care, the results showed a more positive attitude and a higher mean score in the trained group (3.43 ± 0.37 versus 3.21 ± 0.32), the difference being statistically significant (*p* < 0.001).

## 1. Introduction

The COVID-19 pandemic has affected almost 50 million people worldwide and has resulted in more than 1,200,000 deaths [1], with Spain being one of the countries in the world with the highest number of deaths due to COVID-19 of 83.11 per 100,000 people [2]. This has increased the number of patients that Spanish nurses have to care for at the end of their lives. In a situation without precedent, they have been under increased stress, high emotional impact, and negative working conditions, along with concerns about infecting their own relatives [3,4,5,6,7].

The exceptional situation arising from the pandemic has led to the establishment of isolation protocols for patients during their hospitalization, which has made it difficult for their families to accompany and bid them farewell [8,9], both of which are key elements of care [10] to reduce the suffering of the patient [11] as well as to ease the grief and adaptation to the loss of the relatives [12]. It is also a risk factor for the development of pathological grief in the latter [12,13].

Furthermore, training in palliative care provides nurses with greater caring skills, increased safety, and reduced stress [14,15,16,17], thus defining their attitude toward end-of-life care [18].

According to Constantini et.al. [19], the hospice sector is capable of responding flexibly and rapidly to the COVID-19 pandemic. Even these authors affirm that governments must urgently recognize the essential contribution of hospice and palliative care to the COVID-19 pandemic and ensure that these services are integrated into the health care system response.

Palliative care must be extended to all levels of care and services. In this sense, Rosa et.al. [20] recommended the urgent need for palliative care integration throughout critical care settings to support critical care nurses in alleviating suffering during the COVID-2019 pandemic and make recommendations to strengthen nursing capacity to deliver high-quality, person-centered critical care. These authors also claim that nurses should focus on a strategic integration of palliative care, critical care, and ethically based care during times of normalcy and of crisis. Primary palliative care should be provided for each patient and family, and specialist services sought, as appropriate. Nurse educators are encouraged to use these recommendations and resources in their curricula and training.

In addition, increased training of nurses to care for patients at the end-of-life decreases patient suffering [21]. However, training in palliative care for nurses is not compulsory in Spain [22], despite the recommendations of scientific societies [23,24,25] and the evidence shown by several studies [22,25,26]. This means that not all practicing nurses are trained in palliative care.

Along the same line of argument, some studies on fear, death, and the attitudes of professionals toward patient care show medium–high levels of fear to death in those who begin their training [27,28], and how the suffering of others makes them stressful, leading to negative attitudes toward patient care at the end-of-life [29]. These negative attitudes will affect the quality of care provided by nurses as they will develop rejecting or avoiding attitudes [15,30].

This must be kept in mind with how Spain is still the leading country in terms of infection rates and how nursing has acquired the role of a retaining wall in the health system [31].

## 2. Materials and Methods

Our study aims to assess the influence of training on the nurses’ attitudes toward end-of-life care during the COVID-19 pandemic, their stress, motivation, and satisfaction as well as their influence on the accompaniment and farewell of patients as key elements of their care.

A descriptive cross-sectional study was designed for this research aimed at practicing nurses. Out of 250,000 registered nurses according to the Spanish Ministry of Health [32], the sample size was set at 238 participants for a 95% confidence, 3% accuracy, p-ratio = 0.5 (50%), and expected losses of 15%. Given the exceptional circumstances created by the state of alarm in Spain, data collection was conducted online in April and May 2020.

The great added difficulty for the online selection of participants was to carry out probabilistic sampling due to the voluntary self-selection of the participants. To avoid this bias, we used recruitment strategies through natural leaders by means of a snowball technique through social media where the target was nursing professionals, highlighting these participants who adequately represented all the strata of the population under study. These same circumstances have been evidenced in similar studies [8,31]. On the other hand, the possibility of non-response due to the self-selection and exclusion of subjects without access to the Internet is currently a problem diminished by the current digital progress and reinforced by the increase in the use of technology in the nursing community. In the same vein, with the aim of increasing the response rate and decreasing the dropout rate, the research team sent out reminders every 10 days (five in total) that increased the participation of each possible study subject [33].

The proposed inclusion criteria were nursing professionals who cared or had cared for adults in the last days of life with and without COVID-19 in Spain, in the hospital setting (hospital ward, ICU, resuscitation unit) or in the residential setting (nursing homes and public health centers). Home care and pediatric nurses were excluded. Specialist nurses who were working as generalist nurses at the time of the study were not excluded.

Of the 360 responses received, 25 were excluded due to lack of signed informed consent, failure to meet the inclusion criteria, duplicity of registration, or incomplete responses. Questionnaires with identical answers and consecutive time staged were considered as “duplicity of registration”.

Finally, responses from 335 nurses were included, the majority of which were female (86.9%), with an average age of 40.26 years and an average professional practice of 15.79 years, representing the different areas of care. Regarding end-of-life care, 273 nurses cared for COVID-19 patients, 264 cared for non-COVID-19 patients, and 202 cared for both types of patients (see sample Table 1).

### 2.1. Data Collection

Participating nurses were provided with an online self-report questionnaire, each survey was accompanied by an information sheet for incorporation into the study and an informed consent. This questionnaire contained the Attitudes about End-of-life Care score in its Spanish version used by Bermejo et al. (α 0.71) [34], and some socio-demographic questions that included several items related to training, accompaniment, farewell, and the way of caring, based on the literature. The answers were closed or dichotomous or multiple type. Others were on job satisfaction, motivation, and stress; all of these were taken from the study by Hurtado de Mendoza [35] on the perception of psychosocial risks in workers.

The Attitudes about End-of-life Care score includes nine items referring to the professional’s opinion and responsibility in the care of the patient at the end-of-life and their relatives. Responses are Likert-type, ranging from “strongly disagree” (score 1) to “strongly agree” (score 4). The first six items showed favorable attitudes toward patient care when they have high scores. The last three items show unfavorable attitudes and their scores should be reversed beforehand.

Job satisfaction, motivation, and perceived job stress were assessed using a 5-point single-item scale between 1 and 5 (from strongly disagree to strongly agree) for each of the variables: “I am very satisfied in my job”, “I am very motivated in my job”, and “I am very stressed in my job”. These scales demonstrated internal validity and correlation with other validated scales used in the same study.

All responses were collected anonymously without identifying data of the participating subjects and without activation of the automatic collection of information from the respondent. The monitoring of IP address was not activated either, since it does not guarantee the impediment of a subject responding through two different IP addresses. However, different people can use the same intellectual property in a professional context.

### 2.2. Data Analysis

Data were summarized using mean (SD) for numeric variables and absolute frequency (%) for categorical variables. For inferential analysis, the Mann–Whitney U test was used for independent data as well as contingency tables and the chi-square test. For the End-of-Life Care Attitudes and Palliative Care Training scale, we used the average score for a better interpretation of the data. 

Since the studied variables were Likert scales, ordinal regression was performed in order to compare attitude score and education (training in palliative care) including years of professional practice, sex (female), care of COVID-19 patients in their last days, unit where they worked during the COVID-19 pandemic (ICU and Resuscitation vs. Hospitalization and Nursing Home or Public Health Center) as covariable. To analyze the relationship of attitude with training in palliative care, multivariable ordinal regression models were adjusted including the variables: years of professional practice, sex, care of COVID-19 patients in their last days, having encountered deceased patients and to say goodbye to their family for stress and only years of professional practice, and it is not for motivation and satisfaction. *p* values were adjusted using false discovery rate. Frequencies and 95% confidence intervals (CI) were calculated for each variable.

The significance level used was 5%.

Statistical analysis was performed using SPSS software (v20), R software (v4.1.1), and ordinal (v2019.12-10) package.

## 3. Results

The results showed that only 51% of the nurses in the sample had training in palliative care, although all took care of patients toward the end of their life.

Of the nurses, 53.1% were aware of the protocol for accompaniment during the pandemic, with a higher percentage of nurses with palliative care training than those with no training (30.1% compared to 22.4%), with the difference being statistically significant (*p* = 0.003).

Of the 273 nurses caring for COVID-19 patients, 52.4% reported having encountered deceased patients when entering their room, with similar figures between those with and without training (26.7% vs. 25.6%) and no statistical significance (*p* = 0.287). Regarding the nurses who cared for non-COVID-19 patients (264), 46.2% had encountered deceased patients; 30.3% of whom had palliative care training and 15.9% who had not. There were statistically significant differences between them (*p* = 0.013). According to the nurses, only 43.6% of the patients who died from COVID-19 were able to say goodbye to their family, while 76.9% of the non-COVID-19 patients were able to say goodbye to their family (see Table 2). Training was not found to be a significant variable in whether or not such a farewell could or could not be said in either group of patients (*p* = 0.186 and *p* = 0.645, respectively).

Among the nurses, 35.5% stated that the pandemic had changed their way of caring a great deal, with no statistically significant differences in relation to training (*p* = 0.746) and with similar percentages (35.1 and 35.4%).

More than half of the nurses reported agreeing or strongly agreeing with the perception of stress (67.2%), with a higher percentage among those with no training (72.7% vs. 61.4%) (see Table 3).

In the multivariate analysis, training was not decisive in the generation of stress, while finding deceased patients and the care of COVID-19 patients were stressors, with a statistically significant difference in the latter case (*p* = 0.037). Motivation or professional satisfaction of both groups were quite similar (58.4% vs. 59.6% in agreeing or strongly agreeing in satisfaction and 53.2% vs. 49.7% in motivation), although in the multivariate analysis of the sample, the training had a positive influence, which was not statistically significant. Professional experience and being a woman also had a positive influence on both motivation and satisfaction, but only professional experience in satisfaction was significant (*p* = 0.037). In the satisfaction of the nurses, it was also positive to have taken care of COVID-19 patients and to have allowed them to say goodbye to their relatives (see Table 4).

Regarding attitudes toward end-of-life care (Table 5), the results showed a more positive attitude and a higher mean score in the trained group (3.43 ± 0.37 versus 3.21 ± 0.32), the difference being statistically significant (*p* < 0.001).

Focusing on the items of the score scale, we should highlight the positive influence of training in the responsibility of assisting the relatives in dealing with grief after the death of the patient (item 4), with a percentage of 63.7% of those who had training compared to 43.5% of those who did not (*p* = 0.004); in item 5 “Depression can be treated in patients at the end-of-life” with 52% compared to 35.4%*%* (*p* = 0.019); and in item 6 “It is possible to tell the patient the prognosis of terminal illness and still maintain hope” with 24.6% compared to 10.6% (*p* = 0.026). The same applies to the items “Caring for patients at the end-of-life is depressing” (item 7) with a percentage of strongly disagreeing at 49.7% of those who had training compared to 25.5% of those who did not (*p* = 0.001), and “I am afraid of having to deal with the emotional stress of relatives after the death of the patient“ with 35.7% compared to 20.5% (*p* = 0.011).

## 4. Discussion

The training of the nurses in the sample is in line with previous studies on palliative care training in our country, in which only half of the nurses receive such training in their university degree program [22]. The higher percentage among nurses who cared for non-COVID-19 patients may be justified by the higher response from those who routinely cared for end-of-life patients prior to the pandemic, and therefore would have felt the need to undergo training.

Palliative care training provides knowledge about the importance of end-of-life support, so nurses with such training may be more interested in learning about the palliative care protocol established during the pandemic. However, it should be noted that there are low rates of knowledge of the established protocol. Perhaps these figures are due to other factors such as the dissemination of the protocols, high workloads, and staff turnover in different areas of care or recent incorporation into the job. The latter would also explain why the professionals who cared for COVID patients had less training in palliative care, and therefore in end-of-life support, as recent graduates would not have had access to specialized postgraduate training.

The results are notable for the significant number of nurses who found deceased patients alone in their rooms. The restrictive and isolation measures to which patients (both COVID-19 and non-COVID-19) have been subjected during this pandemic have resulted in many of them having to remain alone throughout their hospitalization and in the last moments of their lives. They have been deprived of the right not to die alone, as provided for by the WHO [25], and not allowed to say goodbye to their families [8]. The results on farewell clearly show the influence of isolation protocols [36,37,38,39] on COVID-19 patients without the influence of training. However, it is striking that trained nurses found more deceased non-COVID-19 patients. This may be due to the fact that nearly twice the number of nurses caring for non-COVID-19 patients were trained, while in the COVID-19 group, the number of trained and untrained nurses was very similar.

The unusual situation of the pandemic has changed nursing care at the end-of-life. However, the training should have positively influenced an adaptation with a minor change, as some aspects of it should have remained unchanged due to their importance. Physical contact (even with personal protective equipment), together with allowing the expression of feelings or close monitoring of symptoms for adequate control are some of the cornerstones of palliative care, the knowledge of which should have enabled nurses to adapt care within the established protocol. We agree with other studies [14] showing that training in palliative care is a variable with an important influence on the degree of anxiety and stress of health care professionals. The pandemic has caused an increase in stress levels, although our results showed considerably higher percentages than in other studies [3,4,5]. This upward difference is possibly increased by the anxiety generated by dealing with death as a specific group of professionals caring for patients at the end-of-life.

Understanding how to accompany the dying and their families and how to deal with the emotions caused by death may be key to the level of stress of the professional [28], which once again justifies the need for training in this area [23,24,25,26], together with specific training related to the pandemic. In the same direction, a perception of greater control and less psychological strain (such as that provided by training) leads to less tension and stress, while increasing motivation [35].

The results showed a positive attitude toward end-of-life care as in other studies [34,40,41], despite the pandemic situation. However, our results were only slightly higher in untrained nurses compared to the results of another study [34] conducted in a university population. This increase could be due to the professional experience of the nurses in the sample, as other studies point to more positive attitudes toward death and care of dying patients [41,42].

However, in some of the aspects evaluated by the score scale, there are differences in training, especially in the attention to the grief of family members and in the perception that the care of these patients is depressing. In both cases, training in palliative care teaches the inclusion of the family as a unit to be treated together with the patient and to consider death as a natural process and not as a failure [26,43].

Nevertheless, they may also have been influenced by the special circumstances of the pandemic [5,6,7]. Care for grieving relatives showed lower values (in all nurses) than those indicated by students in training [34], while fear of dealing with the emotional stress of relatives after the death was also lower in those without training, but higher in trained nurses. This decrease may have been affected by the absence of family members due to imposed restrictions [36,37,38,39] and not having had to address such needs in relatives.

### Strengths and Limitations

This study is innovative as it describes how palliative care training is essential during the pandemic, not only for the acquisition of skills to deal with the emotions of facing death and the stress of both patients and families, and of the professionals themselves due to the increased security it provides [15,16,17,19,20]. This supports the influence of training in the possibility of explaining the prognosis of terminal outcome and maintaining hope in a process in which the rapid course of the disease shortens the time of care in its final phase. In addition, a comparison of COVID-19 and non-COVID-19 patients was included to strengthen the influence of the training variable. Another strong feature of this study was the instrument used, validated in Spanish [34], as it allowed us to describe realities associated with the characteristics of death by COVID-19 (loneliness of the process, absence of farewells, etc.). As a limitation, it is important to note that due to the state of alarm decreed in Spain at the time of data collection, we were required to carry out the questionnaire online via social networks.

## 5. Conclusions

Training in palliative care continues to be a key element in the care of people at the end-of-life, increasingly so in times of COVID-19. Specific training in end-of-life accompaniment is needed as well as specialized resources (support from expert staff) by the Clinical Management Units working with COVID-19 patients, in order to provide assistance and comprehensive care in this process. During the COVID-19 pandemic, it is essential for the health system that professionals acquire skills in dealing with death and grief to prevent complicated grief for family members and to reduce post-traumatic stress for the professionals themselves.

## Figures and Tables

**Table 1 ijerph-18-11249-t001:** Socio-demographic data of the sample.

Socio-Demographic Data (N = 335)					
Variables			Mean in Years (SD)		*n* (%)
Sex	Male				44 (13.1)
	Female				291 (86.9)
Age	Male		40.26 (±10.81)	42.57 (±10.94)	
	Female			39.91 (±10.77)	
Years of professional practice	Male		15.79 (±10.71)	18.05 (±10.92)	
	Female			15.45 (±10.75)	
Spanish Autonomous Region	Comunidad Valenciana				67 (20)
	Castilla León				48 (14.3)
	Madrid				34 (10.1)
	Aragón				33 (9.9)
	Islas Baleares				27 (8.1)
	Cataluña				26 (7.8)
	Castilla La Mancha				23 (6.9)
	Andalucía				20 (6)
	Galicia				20 (6)
	Extremadura				14 (4.2)
	País Vasco				7 (2.1)
	Islas Canarias				4 (1.2)
	Murcia				4 (1.2)
	Asturias				4 (1.2)
	Navarra				3 (0.9)
	Cantabria				1 (0.3)
Postgraduate Training	No				156 (46.6)
	Yes				176 (52.5)
	NR/DK				3 (0.9)
Training in Palliative Care	No				161 (48.1)
	Yes				171 (51)
	NR/DK				3 (0.9)
Usual Unit	Surgery				14 (4)
	Geriatrics				5 (1.4)
	Oncology and Hematology				22 (6.4)
	Urology and Nephrology				11 (3.2)
	Internal Medicine and Infectious Diseases				56 (16.2)
	ICU and Resuscitation				50 (14.5)
	Emergency				26 (7.5)
	Palliative Care Unit				35 (10.1)
	Pneumology				22 (6.4)
	Gastrointestinal				6 (1.7)
	Operating Theatre				10 (2.9)
	Traumatology				7 (2)
	Cardiology				6 (1.7)
	Gynecology and Pediatrics				5 (1.4)
	Neurology				5 (1.4)
	Nursing Home or Public Health Centre				37 (10.7)
	Undefined Hospitalization				15 (4.3)
	Others				14 (4)
Unit during COVID-19 pandemic	Specialized Care	Hospitalization			224 (66.9)
		ICU			52 (15.5)
		Resuscitation (restructures due to COVID-19)			18 (5.4)
	Nursing Home or Public Health Centre				41 (12.2)
Care of patients during COVID-19 pandemic	Care of COVID-19 patients in their last days				273 (81.49)
	Care of non-COVID-19 patients in their last days				264 (78.8)

Abbreviations: Standard Deviation (SD), No response/Do Not Know (NR/DK), Intensive Care Unit (ICU).

**Table 2 ijerph-18-11249-t002:** Farewell to COVID-19 and non-COVID-19 patients and palliative care training for the nurses who cared for them.

		Yes	No	NR/DK	Total	*p*-Value
		*n* (%)	*n* (%)	*n* (%)	*n* (%)	
	Allowed farewell to COVID-19 patients					
Training in Palliative Care	Yes	60 (22)	64 (23.4)	5 (1.8)	129 (47.3)	0.186
	No	58 (21.2)	71 (26)	12 (4.4)	141 (51.6)	
	NR/DK	1 (0.4)	1 (0.4)	1 (0.4)	3 (1.1)	
	Total	119 (43.6)	136 (49.8)	18 (6.6)	273 (100)	
	Allowed farewell to non-COVID-19 patients					
Training in Palliative Care	Yes	114 (43.2)	29 (11)	6 (2.3)	149 (56.4)	0.645
	No	86 (32.6)	18 (6.8)	8 (3)	112 (42.4)	
	NR/DK	3 (1.1)	0 (0)	0 (0)	3 (1.1)	
	Total	203 (76.9)	47 (17.8)	14 (5.3)	264 (100)	

Abbreviations: No Response/Do Not Know (NR/DK).

**Table 3 ijerph-18-11249-t003:** Satisfaction, stress, and motivation based on their training in palliative care.

	Strongly Disagree (1)	Disagree (2)	Neither Disagree nor Agree (3)	Agree (4)	Strongly Agree (5)	
Training in Palliative Care	*n* (%)	*n* (%)	*n* (%)	*n* (%)	*n* (%)	Total *n* (%)
	Stress due to professional practice: “I am very stressed in my job”					
Yes	22 (12.9)	25 (14.6)	19 (11.1)	58 (33.9)	47 (27.5)	171 (100)
No	11 (6.8)	14 (8.7)	19 (11.8)	66 (41)	51 (31.7)	161 (100)
NR/DK	0 (0)	0 (0)	0 (0)	3 (100)	0 (0)	3 (100)
	Motivation in professional practice: “I am very motivated in my job”					
Yes	18 (10.5)	32 (18.7)	30 (17.5)	57 (33.3)	34 (19.9)	171 (100)
No	17 (10.6)	24 (14.9)	40 (24.8)	56 (34.8)	24 (14.9)	161 (100)
NR/DK	0 (0)	0 (0)	2 (66,6)	0 (0)	1 (33.3)	3 (100)
	Satisfaction in professional practice: “I am very satisfied in my job”					
Yes	13 (7.6)	35 (20.8)	23 (13.5)	63 (36.8)	37 (21.6)	171 (100)
No	14 (8.7)	31 (19.3)	20 (12.4)	67 (41.6)	29 (18)	161 (100)
NR/DK	0 (0)	1 (33.3)	1 (33.3)	0 (0)	1 (33.3)	3 (100)

Abbreviations: Standard Deviation (SD), No Response/Do Not Know (NR/DK).

**Table 4 ijerph-18-11249-t004:** Satisfaction, stress, and motivation based on their training in palliative care. Ordinal multivariable regression.

	Odds Ratio			Confidence Intervals (Lower95–Upper95)			*p*-Value		
Variables	Stress Due to Professional Practice	Motivation in Professional Practice	Satisfaction in Professional Practice	Stress due to Professional Practice	Motivation in Professional Practice	Satisfaction in Professional Practice	Stress due to Professional Practice	Motivation in Professional Practice	Satisfaction in Professional Practice
Training in Palliative Care	0.752	1.131	1.214	0.498–1.134	0.76–1.685	0.808–1.827	0.174	0.545	0.351
Years of professional practice	0.984	0.996	0.98	0.965–1.003	0.978–1.014	0.962–0999	0.091	0.646	0.037
Sex: Female	0.754	1.019	1.058	0.43–1.314	0.591–1.753	0.594–1.88	0.32	0.945	0.847
Care of COVID-19 patients in their last days	1.702		1.365	1.029–2.816		0.819–2.277	0.038		0.232
Having encountered deceased patients	1.217		0.864	0.814–1831		0.578–1.29	0.34		0.474
To say goodbye to their family	0.705		1.322	0.455–1.088		0.867–2.02	0.116		0.195

Análisis (estr_s_pr_cticaprof)~formaci_ncp + a_osprofesi_n + sexo + cuida_COVID + fallecidos + despedida, data = datos [!datos$formaci_ncp%in% “NS/NC”]), motivaci_n_pr_cticaprof)~formaci_ncp + a_osprofesi_n + sexo, data = datos [!datos$formaci_ncp%in% “NS/NC”]) satisfacci_n_pr_cticaprof) ~ formaci_ncp + a_osprofesi_n + sexo + cuida_COVID + fallecidos + despedida, data = datos [!datos$formaci_ncp%in% “NS/NC”]).

**Table 5 ijerph-18-11249-t005:** Score of Attitudes about End-of-Life Care and Training in Palliative Care.

	Attitudes about End-of-Life Care Score (N = 335)								
Item Value *			Sd (1)	D (2)	A (3)	SA (4)			
		Mean ± SD	*n* (%)	*n* (%)	*n* (%)	*n* (%)	Total *n* (%)	OD [CI]	*p*-Value
	1: Psychological suffering can be as hard as physical suffering.								
Training in Palliative Care	Yes	3.92 ± 0.343	1 (0.6)	1 (0.6)	8 (4.7)	161 (94.2)	171 (100)	1.203 [0.444–3.313]	0.715
	No	3.94 ± 0.242	0 (0)	0 (0)	10 (6.2)	151 (93.8)	161 (100)		
	NR/DK	4 ± 0	0 (0)	0 (0)	0 (0)	3 (100)	3 (100)		
	2: Health care professionals have a responsibility to help patients at the end of their lives.								
Training in Palliative Care	Yes	3.95 ± 0.212	0 (0)	0 (0)	8 (4.7)	163 (95.3)	171 (100)	1.228 [0.458–3.419]	0.685
	No	3.93 ± 0.286	0 (0)	1 (0.6)	10 (6.2)	150 (93.2)	161 (100)		
	NR/DK	4 ± 0	0 (0)	0 (0)	0 (0)	3 (100)	3 (100)		
	3: Health care professionals have a responsibility to help patients prepare for death.								
Training in Palliative Care	Yes	3.79 ± 0.475	1 (0.6)	2 (1.2)	29 (17)	139 (81.3)	171 (100)	1.687 [0.981–2.931]	0.06
	No	3.68 ± 0.529	0 (0)	5 (3.1)	41 (25.5)	115 (71.4)	161 (100)		
	NR/DK	4 ± 0	0 (0)	0 (0)	0 (0)	3 (100)	3 (100)		
	4: Health care professionals have a responsibility to support the grieving family members after the death of the patient.								
Training in Palliative Care	Yes	3.47 ± 0.821	7 (4.1)	15 (8.8)	40 (23.4)	109 (63.7)	171 (100)	1.936 [1.24–3.04]	0.004
	No	3.27 ± 0.740	1 (0)	25 (15.5)	65 (40.4)	70 (43.5)	161 (100)		
	NR/DK	4 ± 0	0 (0)	0 (0)	0 (0)	3 (100)	3 (100)		
	5: Depression can be treated in patients at the end-of-life.								
Training in Palliative Care	Yes	3.39 ± 0.714	1 (0.6)	20 (11.7)	61 (35.7)	89 (52)	171 (100)	1.693 [1.092–2.634]	0.019
	No	3.17 ± 0.729	2 (1.2)	25 (15.5)	77 (47.8)	57 (35.4)	161 (100)		
	NR/DK	3.33 ± 0.577	0 (0)	0 (0)	2 (66.7)	1 (33.3)	3 (100)		
	6: It is possible to explain the terminal prognosis to the patient and still maintain hope.								
Training in Palliative Care	Yes	2.88 ± 0.839	9 (5.3)	44 (25.7)	76 (44.4)	42 (24.6)	171 (100)	1.624 [1.061–2.494]	0.026
	No	2.60 ± 0.808	16 (9.9)	49 (30.4)	79 (49.1)	17 (10.6)	161 (100)		
	NR/DK	2.33 ± 0.577	0 (0)	2 (66.7)	1 (33.3)	0 (0)	3 (100)		
Item Value *			Sd (4)	D (3)	A (2)	SA (1)			
	7: Caring for patients at the end-of-life is depressing.								
Training in Palliative Care	Yes	3.19 ±0.958	85 (49.7)	45 (26.3)	29 (17)	12 (7)	171 (100)	0.48 [0.313–0.733]	0.001
	No	2.80 ± 0.934	41 (25.5)	63 (39.1)	41 (25.5)	16 (9.9)	161 (100)		
	NR/DK	3.67 ± 0.577	2 (66.7)	1 (33.3)	0 (0)	0 (0)	3 (100)		
	8: I feel guilty after the patient’s death.								
Training in Palliative Care	Yes	3.43 ± 0.728	96 (56.1)	55 (32.2)	18 (10.5)	2 (1.2)	171 (100)	0.724 [0.465–1.126]	0.153
	No	3.27 ± 0.873	83 (51.6)	45 (28)	27 (16.8)	6 (3.7)	161 (100)		
	NR/DK	2.67 ± 0.577	0 (0)	2 (66.7)	1 (33.3)	0 (0)	3 (100)		
	9: I am afraid of having to deal with the emotional stress of relatives after the death of the patient.								
Training in Palliative Care	Yes	2.87 ± 1.032	61 (35.7)	47 (27.5)	43 (25.1)	20 (11.7)	171 (100)	0.582 [0.383–0.883]	0.011
	No	2.48 ± 1.031	33 (20.5)	44 (27.3)	52 (32.3)	32 (19.9)	161 (100)		
	NR/DK	2.33 ± 1.528	1 (33.3)	0 (0)	1 (33.3)	1 (33.3)	3 (100)		

Abbreviations: Standard Deviation (SD), Odds Ratio (OD), Confidence Intervals (CI), Strongly disagree (Sd), Disagree (D), Agree (A), Strongly Agree (SA), No response/Do Not Know (NR/DK). * The items 1,2,3,4,5 and 6 show favorable attitudes and the last three items show unfavorable attitudes (7,8,9). Clm (formula = ordered (ayuda_preparaci_nmuerte)~formaci_ncp + a_osprofesi_n + sexo + cuida_COVID + ambito_pandemia_cat, data = datos [!datos$formaci_ncp %in% “NS/NC”]).

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
