# Peer review of "Influence of Palliative Care Training on Nurses’ Attitudes towards End-of-Life Care during the COVID-19 Pandemic in Spain"

_ijerph, 2021, doi:10.3390/ijerph182111249_

Round 1
Reviewer 1 Report
While the minor changes to the manuscript have modestly improved it, the lack of any additional empirical work makes it difficult to believe these results as being causal in any way.
Author Response
Reviewer response: Thank you very much for your very pertinent comments, thanks to them, we have included your best explanation.
Our intention is not to establish causality, but the relationship of the formation with the items of the scale that presented statistically significant differences. As well as that training is essential or a key element for the care of palliative patients.
The mean of the Likert scales was calculated for a better interpretation of the results, following the steps of the authors of the Validation in Spanish of the Attitudes about End-of-Life Care and Training in Palliative Carescale. This allowed us to compare our results with yours.
Reviewer 2 Report
The authors have taken on board the observations made and made the required changes. The statistical analysis is now more robust and supports the conclusions.
Author Response
Thank you very much for your words that have caused us so much encouragement to the research team, it has been a hard work during the pandemic that has been rewarded with your recognition. Thanks a lot.
Reviewer 3 Report
Thank you for addressing most of my comments.
I guess authors want to refer to multivariate regression analysis of ordinal data instead of Ordinal multivariable regression in the text. Authors should better extend this sensitive point in data analysis to allow replications.
Author Response
Thank you very much for your coherent methodological advice that has allowed us to include in the text a small paragraph that will make the reader more understandable. Our analysis is an ordinal regression model which is best suited for analyzing ordinal variables (such as Likert scales).
Again I reiterate my gratitude
Round 2
Reviewer 3 Report
Thank you for addessing my suggetsions
This manuscript is a resubmission of an earlier submission. The following is a list of the peer review reports and author responses from that submission.
Round 1
Reviewer 1 Report
This paper tests to see whether training in palliative care affects the manner in which nurses provide care and the attitudes towards end of life care, particularly during the COVID period.
The researchers do find some interesting differences between nurses who have had palliative care training and those who have not. In particular, nurses who have such training have a more balanced and hopeful view of death than ones who have not.
The main problem with such an observational study is that there is no attempt to deal with the self-selection problem that exists. It is likely the case that nurses who are likely to have more hopeful and positive views toward end of life care are the ones who would most likely seek to get training in palliative care. As this is not a randomized experiment or even quasi or natural experiment, it is difficult to realiably believe these to be causal results. At the very least, one could conduct multiple linear regressions that control for characteristics of the nurses such as age and experience, geographic location, other general attitudes towards life and nursing. Without these controls, it is hard to place much credence on these results. The authors do not discuss the issue of selection bias and how that affects the results.
Minor points:
Tables 3 and 4 are presented inconsistently, as the Yes appears above the No group (for having training in palliative care) in Table 3, but the No appears above the Yes group in Table 4. This should be the same across both tables.
Also, in Table 4, the scaling changes for questions 7,8, and 9. I understand the reasoning for this because these questions are worded differently than for questions 1-6, but the table should show a break between these sets of questions so the different scaling is more prominent and easy for the reader to see.
Reviewer 2 Report
The article aims to evaluate the effect of training on nurses' attitudes towards end-of-life care during the pandemic.
The work shows several shortcomings from a methodological point of view.
First of all, in order to evaluate the effect of training, a longitudinal design should be used: the authors, due to the pandemic, preferred to adopt a transversal design whose effectiveness is based on strict control of many intervening variables (type of training undergone, length of service, level of education, etc.) The article does not mention the use of any sampling or post hoc statistical techniques to control the effect of these variables.
Secondly, the online questionnaire with which the data were collected is not described at all: the wording of the questions and the type of scales used are not known.
The third methodological defect is the presentation of the data in tables that are too dense with information and difficult to read.
And finally, the authors report statistical probabilities without defining from what type of data and with what type of analysis they were obtained.
Therefore, the conclusions, which can be shared on the basis of common sense, do not have any scientific characteristics.
Reviewer 3 Report
This is an interesting manuscript on palliative care training on nurses' attitudes towards end-of-life care during the COVID-19. An extra value is the sample size from different regions in the country.
Even if I see promise here, many issues should be addressed (particularly on the method and results presentation):
The introduction is veru short. There is much to be said regarding this topic. Please see:
Costantini, M., Sleeman, K. E., Peruselli, C., & Higginson, I. J. (2020). Response and role of palliative care during the COVID-19 pandemic: a national telephone survey of hospices in Italy. Palliative medicine, 34(7), 889-895.
Rosa, W. E., Ferrell, B. R., & Wiencek, C. (2020). Increasing critical care nurse engagement of palliative care during the COVID-19 pandemic. Critical care nurse, 40(6), e28-e36.
I also suggest to include recent literature on grief facing Covid-19 outbreak:
Murphy, M., & Moret-Tatay, C. (2021). Personality and attitudes confronting death awareness during the COVID-19 outbreak in Italy and Spain. Frontiers in Psychiatry, 12, 74.
Pérez-Mengual, N., Aragonés-Barbera, I., Moret-Tatay, C., & Moliner-Albero, A. R. (2021). The Relationship of Fear of Death Between Neuroticism and Anxiety During the Covid-19 Pandemic. Frontiers in Psychiatry, 12, 501.
With regards to methodology, authors should present the psychometric properties of the scales under study.
More information about the passing of the questionnaire has to be provided, how did they ensure that it was anonymous? if it is anonymous how could the participants make use of their right to withdraw the data? Were IPs controlled? How did you deal with duplicate data?
They are only using likert scales as ordinals variables, when they can be used and much more information can be extracted than they provide.
The p-values must be accompanied by the effect size.